# Patterns of Glycemic Variability During a Diabetes Self-Management Educational Program

**DOI:** 10.3390/medsci7030052

**Published:** 2019-03-25

**Authors:** Ankur Joshi, Arun Mitra, Nikhat Anjum, Neelesh Shrivastava, Sagar Khadanga, Abhijit Pakhare, Rajnish Joshi

**Affiliations:** 1Department of Community and Family Medicine, All India Institute of Medical Sciences (AIIMS), Bhopal 462020 India; ankur.cfm@aiimsbhopal.edu.in (A.J.); dr.arunmitra@gmail.com (A.M.); abhijit.cfm@aiimsbhopal.edu.in (A.P.); 2Hospital Services, All India Institute of Medical Sciences, Bhopal 462020, India; nutritionistnikhat@gmail.com; 3Department of Medicine, All India Institute of Medical Sciences, Bhopal 462020, India; neeleshstv@gmail.com (N.S.); sagar.genmed@aiimsbhopal.edu.in (S.K.)

**Keywords:** glycemic variability, MAGE, CONGA, MODD, HBGI, LBGI

## Abstract

Background: Variations in blood glucose levels over a given time interval is termed as glycemic variability (GV). Higher GV is associated with higher diabetes-related complications. The current study was done with the aim of detecting the sensitivity of various GV indices among individuals with type 2 diabetes mellitus of different glycemic control status. Methods: We performed a longitudinal study among individuals with type 2 diabetes mellitus (T2DM) who were participating in a two-week diabetes self-management education (DSME) program. Participants were categorized by their HbA1c as poor (≥8%), acceptable (7%–8%), and optimal control (<7%). Continuous glucose monitoring (CGM) sensors recorded interstitial glucose every 15 min from day 1. The evaluated GV measures include standard deviation (SD), coefficient of variation (CV), mean amplitude of glycemic excursion (MAGE), continuous overlapping net glycemic action (CONGA), mean of daily difference for inter-day variation (MODD), high blood glucose index (HBGI), and low blood glucose index (LBGI). Results: A total of 41 study participants with 46347 CGM values were available for analysis. Of 41 participants, 20 (48.7%) were in the poor, 10 (24.3%) in the acceptable, and 11 (26.8%) in the optimal control group. The GV indices (SD; CV; MODD; MAGE; CONGA; HBGI) of poorly controlled (77.43; 38.02; 45.82; 216.63; 14.10; 16.62) were higher than acceptable (50.02; 39.32; 30.79; 138.01; 8.87; 5.56) and optimal (34.15; 29.46; 24.56; 126.15; 8.67; 3.13) control group. Glycemic variability was reduced in the poorly and acceptably controlled groups by the end of the 2-week period. There was a rise in LBGI in the optimally controlled group, indicating pitfalls of tight glycemic control. Conclusion: Indices of glycemic variability are useful complements, and changes in it can be demonstrated within short periods.

## 1. Introduction

The prevalence of diabetes mellitus is rising, with 1 in 11 adults worldwide affected, and more than 90% being type 2 diabetes mellitus (T2DM) [1]. Achieving optimal glycemic control is necessary to prevent diabetes-related complications [2] and various pharmacologic and non-pharmacological interventions are essential components of diabetes care [3]. Studies show that about half of all patients with advanced T2DM fail to achieve glycemic control, largely due to post-prandial hyperglycemia [4]. Fluctuations in blood glucose levels over a given interval of time is referred to as glycemic variability (GV) [5]. GV has, not only been associated with poor glycemic control, but also with a poor quality of life and increased risk of diabetes-related complications [6]. Recent studies also implicate GV as an independent risk factor for total mortality and death due to cardiovascular disease in both type 1 and type 2 diabetes [5,7,8]. On the contrary, a post hoc analysis (Diabetes Control and Complications Trial) recognized no contribution of short-term GV measures with complications after adjusting for mean glucose values [9]. However, this should be read with two caveats. Firstly, there is no consensus about the operational construct of short- and long-term GV measures. The second caveat is related to the extent of capturing the true the mean glucose value by 7-point glucose estimation method.

Numerous metrices for glycemic variability have been reported [10,11,12,13,14,15,16]. These include within-day glycemic variability (standard deviation (SD), coefficient of variation (CV), mean amplitude of glycemic excursion (MAGE) [10]), continuous overlapping net glycemic action (CONGA*n*) [16] at n-hour intervals), and between-day (mean of daily differences (MODD) [11,17] glycemic variability. High and low blood glucose indices (LBGI, HBGI) [15] depict the risk of hyperglycemia and hypoglycemia. Table 1 depicts the definition and interpretation of common metrices. With the advent of continuous glucose monitoring (CGM) technology, recording of interstitial glucose values at intervals as short as 5 min have become possible and various measures of GV [18,19,20]. With a plethora of such measures described, we do not know if these measures are differentially affected while we attempt to achieve optimal glycemic control in a spectrum of patients with poorly controlled diabetes mellitus [21].

Diabetes self-management education (DSME) to promote higher physical activity levels, dietary modifications, and optimizing drug therapy targeted at patients with poorly controlled diabetes mellitus are likely to achieve improved short- and intermediate-term glycemic control [22,23,24]. In this context, the current study was done with the aim of detecting the sensitivity of different GV measures to capture this gradient of change in glycemic status and, further, to detect the extent and direction of agreements amongst different GV measures.

## 2. Materials and Methods

### 2.1. Design

We designed a longitudinal study of two-week duration to understand glycemic variability amongst patients with type 2 diabetes mellitus. All participants provided written informed consent prior to initiation of study procedures. This study was approved by the Institutional Human Ethics Committee of AIIMS Bhopal on 16 December 2016 with the reference number IHEC-LOP/2017/EF0036.

### 2.2. Setting

All India Institute of Medical Sciences, Bhopal is a tertiary care hospital in central India. Patients with type 2 diabetes mellitus are followed up in the diabetes clinic in the Department of Medicine, which maintains contact details, appointment schedules, and a record of sequential glycosylated hemoglobin (HbA1c) levels. We designed a two-week DSME program to understand the effect of dietary interventions, where we were assessing glycemic control using CGM-based measurements. The current study was conducted in the setting of such a program.

### 2.3. Participants

We identified adults (aged more than 18 years) having type 2 diabetes and a HbA1c of more than 7% on two previous visits over the past one year from the records of the diabetes clinic. All such 379 participants were first telephonically contacted to identify those who lived within a 5–10 km radius of the hospital and were willing to enroll for a two-week DSME program on pre-specified dates. All potentially eligible participants were invited in the institute for a pre-screening visit. The purpose of this visit was to appraise them synchronously in details about the DSME schedule and pre-conditions to adhere with this schedule. The persons who were in agreement to follow the schedule for DSME were enrolled, and a venous blood sample was drawn to obtain a baseline HbA1c value. Of the 84 individuals who came for the pre-screening visit, 48 agreed to participate.

### 2.4. Procedures

Multi-modal DSME programs were conducted in two batches for the willing participants between April and June 2017. A key objective of this program was to explore the feasibility, and the effect of intensive education and low glycemic index breakfast (served in some of the sessions) to improve glycemic control. Each program batch had 24 participants and was of a 14-day duration. Participants were engaged all 14 days between 6 am and 8 am in the morning. On the first day, a questionnaire was administered to all participants, and information about their demography, comorbidities, known complications, and drug therapy was obtained. Further, participants also self-reported their engagement in diabetes-related self-care activities and performed an assessment of their quality of life (QoL). Thereafter, the study physicians (RJ, AP, and SK) reviewed their previous prescriptions. Fasting blood samples and urine samples were collected on the fifth day of the program to test for serum creatinine, lipids, and to obtain urine albumin to creatinine ratio. Participants were engaged in activity sessions on different days relating to knowledge enhancement (related to disease, its complications, and control measures), physical activity promotion (a 30-min, 3-km brisk walk on all days, yoga sessions), dietary modification (meal planning, maintaining a food diary, and introducing low-glycemic index breakfast), coping strategies (stress reduction), foot care, and drug optimization. These activity sessions (Table 2) were conducted on different days by a team of physicians, nutritionist, physical activity trainer, yoga therapist, and a clinical psychologist. The duration of activity sessions was between 15–30 min.

A continuous glucose monitoring (CGM) sensor was placed on the upper arm or back (Free style Libre-Pro, Abbott Laboratories. Abbott Park, Illinois, USA) of all participants on the first day of the workshop, and daily retrospective review of readings was done using a reader. This sensor records interstitial glucose (IG) value after every 15 min, for a total of 14 days. These IG values correspond to the blood glucose (BG) values. Participants were provided individualized feedback based on CGM readings, food diary, and activity records. The sensors were removed after 14 days or earlier if not tolerated. CGM sensors were provided by the institute free of charge to the study participants.

### 2.5. Statistical Analysis

All the participants were classified in three glycemic control categories, based on HbA1c value obtained at the pre-screening visit—less than 7% as optimal control, 7%–7.9% as acceptable control, and 8% or higher as poor control [25,26]. Glucose values stored in the sensor were downloaded using Free-style Libre-Pro software in MS-Excel format. The time-stamped series of CGM data for each individual was segregated into respective glycemic control categories. We analyzed the obtained cumulative data in each category for its value, rate, and direction of change. Since the CGMS device was inserted and removed at different clock-hour times, we censored the data on the day of insertion and removal. Further, all measures were summarized for a 24-h period. To evaluate changes in variability we considered values between day 1 and 2 as a baseline and compared these with those between days 7 and 8 (mid-program), and days 12 and 13 (end program). We summarized other baseline measures by standard measures of central tendency and dispersion.

Glycemic variability measures were calculated as per definitions provided in Table 1. A graphical representation of the percentage change from baseline in GV measures at the midpoint and at the end of two weeks was done. Lag plots of a few patients from each group were used to visualize the dynamic behavior of the system under scrutiny. Each point of the plot had two coordinates—BG value at time *t* on *x*-axis and BG value at the time (t + 1 h) on y-axis. The difference (Δ*y*) of coordinate represented the BG rate of change in 1 h. We plotted 3 such graphs (one for each group) at day 0, day 7, and day 14. A gradual, more concerted graph was perceived as a marker of system stability and the scattered plot was perceived as an irregular system. A subjective comment of the trend of dynamic regularity was made. The data analysis and graphical analysis was done using R version 3.4 [27]. Details regarding the cleaning of data and computation of GV indices are provided in the Appendix A.

## 3. Results

A total of 54,524 such values from 46 participants were available for analysis. Out of these, five participants were excluded as they did not have recordings for the entire duration of the program due to early sensor removal. Of the 41 participants included in the analysis, 20 (48.7%) had poor glycemic control, 10 (24.3%) had acceptable control, and 11 (26.8%) had optimal glycemic control. The baseline characteristics of all participants are provided in Table 3.

As compared to the baseline, there was a fall in the mean BG values from the baseline (B) to the end of intervention (I) in all three groups (poorly controlled group: 203.6 mg/dl (B) to 144.62 mg/dl (I); acceptable control group: 127.2 mg/dl (B) to 102.2 mg/dl (I); and optimal control group: 115.9 mg/dl (B) to 98.21 mg/dl(I)). A reduction in glycemic variability indices like SD, CV, MODD, MAGE, and CONGA was also observed (Table 4).

These indices had a similar declining pattern in poorly controlled and acceptably control sub-groups as the minimal change in optimally controlled group (Figure 1). LBGI increased markedly (2.3–5.95) at the end of the 2-week program among the optimally controlled group, while it remained similar in the acceptable control and poor control subgroups. HBGI, a measure of duration spent in hyperglycemia, got halved from baseline among the poor control (16.62–7.16) and acceptable control (5.56–1.34) subgroups.

The subgroup of participants with acceptable glycemic control had a larger decline in variability, as compared to those with poor or optimally controlled (Figure 1). The predominant change in optimally controlled was a reduction in glycemic variability towards the end of the program. Those with acceptable control achieved a larger reduction in variability, with a tendency to match glycemic status as the optimally controlled. Those with poor control had a major shift in terms of glycemic control, and an intermediate shift in glycemic variability. Figure 2 presents a lag plot, in which glucose value at the time (t) is plotted against the glucose values at the fourth time-point (60-minute lag). This fourth-order lag plot visually depicts that in all groups, DSME led to a more concentric geometrical ellipsoid at the end of the intervention. The relative change in shape was more visible amongst poor control compared to the optimal control group. This pattern suggests the single-cycle sinusoidal model with the presence of outliers. The outliers were more visible amongst the poor control group. Both facts in cohesion may suggest a mix of generally responsive and few resistant (to DSME) participants in the poor control group. The more concentric geometrical shape which follows the diagonal indicates the presence of stronger auto-correlations as the DSME proceeds. Clinically, this may be translated into less dramatic fluctuations of blood sugar values and seemingly predictable blood sugar values at the end of DSME compared to the beginning.

## 4. Discussion

In the current study, we described changes in glycemic variability and glycemic control amongst individuals with T2DM, during a DSME program. As anticipated, we noticed improved glycemic control, as well as reduced glycemic variability in all the subgroups (optimal, acceptable, and poor glycemic control at baseline). The performance of glycemic variability indices that are based on measures of central tendency (e.g., MAGE, CONGA, and MODD) was similar. Indices that are influenced by extreme values (e.g., LBGI and HBGI) had a differential pattern.

The ability to estimate average blood glucose by a single test such as HbA1c has revolutionized diabetes care, and technologies have now enabled us to obtain this measure at point-of-care. While this measure of central tendency over a period of three months is pivotal, a measure of dispersion of blood glucose values has eluded diabetes care providers. Strides in continuous glucose monitoring technology have the potential to provide a summary measure of dispersion or glycemic variability [5,12,14]. Extreme changes in blood glucose level from fasting to a fed state have well-established health consequences, but computational complexity and absence of benchmarks for optimal glycemic variability measures and levels are key barriers for their wider clinical application [5,7,8]. The current study helps to understand variability measures and their interpretation.

Individuals with T2DM who are optimally controlled at baseline are an acceptable benchmark to define achievable limits of glycemic variability. The range of glycemic variability measures from end to start of the DSME program in this subgroup in our study was 23.14–24.56 for MODD, 111.55–126.15 for MAGE, and 7.61–8.67 for CONGA. Thus, intermediate values of 24 for MODD, 119 for MAGE, and 8 for CONGA could be acceptable benchmarks for glycemic variability. As compared to these benchmarks, the poorly controlled subgroup in our study achieved MODD of 32, MAGE of 154, and CONGA of 8.4 towards the end of the two-week DSME program. In terms of proportion, this subgroup was 25%, 22%, and 4.7% away from benchmark MODD, MAGE, and CONGA.

In the family of glycemic variability indices, MODD has its unique place, as it can measure inter-day variations [12,17]. Originally described as a measure of difference on two successive days for a similar volume of diet and therapy, this measure negates the effect of diurnal variation. With the availability of CGM data, we can estimate MODD for every value 15-min apart at time (t and t + 24 h). In comparison, MAGE measures intra-day variation. This variability measure ignores minor fluctuations (within 1 SD) and averages major deflections (beyond 1 SD) [5,10,12]. Higher MAGE indicates a greater fluctuation during the day. CONGA, like MAGE, also reflects intra-day variation [16]. The two values are different, as MAGE reflects mean and CONGA reflects standard deviation. Further, CONGA does not ignore minor fluctuations. In this regard, MAGE is more intuitive to the clinician, as it reflects average blood glucose. CONGA is more intuitive to a statistician as it summates standard deviation.

The gradients of decline in these three measures (MODD, MAGE, and CONGA) are analogous. The gradient is more in those with poor or acceptable control as compared to optimal control, as the greater the dispersion the more there is opportunity for change. As the educational intervention proceeds, it has a cumulative effect on glycemic fluctuations and dispersion of both inter-day and intra-day values will be reduced. Thus, a mutual comparison of these variability measures shows high mathematical concordance, and any one intuitive measure would be reasonable.

Hypoglycemia is a trade-off for optimal glycemic control. Risk of hypoglycemia is reflected by LBGI— the higher the value, the greater the risk [12,15]. This value increased from 2.30–5.95 in those optimally controlled at baseline but declined from 4.08–3.47 and fluctuated between 2.14 and 4.06 in those with acceptable and poor control, respectively. A mid-intervention value of 4 in the acceptable control subgroup could be a reasonable benchmark for LBGI. Higher values will reflect an increased risk for hypoglycemic events. Compared to this benchmark, those with optimal control had 48% higher LBGI at the end of the intervention, suggesting the need for de-escalation of drug therapy if adherence to educational measures was sustained. In this regard, LBGI has utility as a summary intra-day measure of hypoglycemia. In contrast, HBGI reflects the risk of hyperglycemia and was higher in poorly controlled and acceptably controlled than optimally controlled. A notable observation with respect to HBGI is that it got markedly reduced in poorly controlled (16.62–7.16) and in acceptably controlled (5.56–1.34) over the 2-week DSME program, thereby indicating its sensitivity in assessing glycemic control. This fact may also be seen in the background that, theoretically and computationally, glycemic safety ranges from alleged mid values on both sides are not symmetrical and highly narrowed towards lower glycemic values. Moreover, in long-standing diabetes, the clinical presentation of hypoglycemia and threshold values for manifestations show significant variations inter-individually, which seems to be a complex non-linear interaction of decreased responsiveness of the neuro-sympathomimetic system and increased receptor thresholds. Hence, a large safety margin, especially for the optimal control group, is always warranted.

Several other structured predominantly non-pharmacological interventions, like an introduction to mobile diabetic self-care system and real-time continuous glucose monitoring system, have also reported the decrement in GV measures following interventions [28,29]. All these interventions were of longer duration compared to the present intervention. This study shows that GV measures are sensitive enough to capture the fluctuations in blood glucose values as a result of short-term intensive interventions. Another study with the objective of checking the response of bean and rice meals on postprandial glycemic status also used some indirect methods for GV (area under curve) and showed a favorable response of dietary intervention [30].

Variability is the key feature of all biological systems and is an intricate reflection of endogenous (e.g., insulin reserve, stage of disease) and exogenous (e.g., drugs, diet, physical activity) influences [5,20]. The strength of our study is the use of CGM data to describe different variability indices in the same population. This was done in the context of a DSME program, to demonstrate potential limits of glycemic variability indices. A key limitation of our approach is a small sample size in each of the subgroups. Further, the numbers were too small for us to characterize variability amongst individuals with different oral hypoglycemic agents or insulin. Optimization of drug therapy was an integral part of the DSME program, and hence, is a greater reflection of multi-modal change rather than of any initiative. The study was not designed to compare changes glycemic variability with those who would not have enrolled in the program, hence we are unable to make any such comparisons.

## 5. Conclusions

To conclude, glycemic variability indices were measured and have shown changes over a short period of time and, thus, could be used in clinical practice to assess glucose control status. Various measures correlate with each other, and hence, a most intuitive measure of change (such as MAGE or MODD and LBGI) may be used for clinical application. The utility of individual measures in glycemic control also needs to be evaluated in terms of its ability to reduce variability.

## Figures and Tables

**Figure 1 medsci-07-00052-f001:**
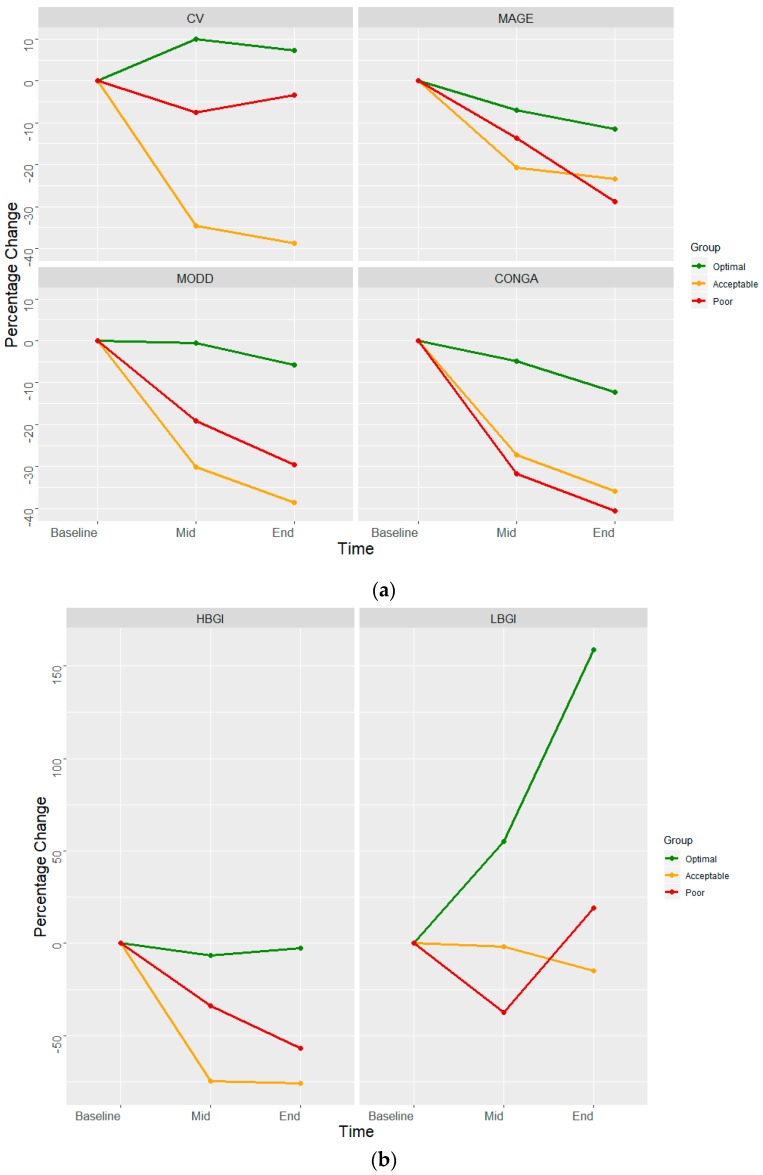
Percentage change in glycemic variability indices. (**a**) coefficient of variation (CV), mean amplitude of glycemic excursion (MAGE), mean of daily difference for inter-day variation (MODD), and continuous overlapping net glycemic action (CONGA). (**b**) High blood glucose index (HBGI) and low blood glucose index (LBGI) from baseline among various glycemic control categories (risk plot).

**Figure 2 medsci-07-00052-f002:**
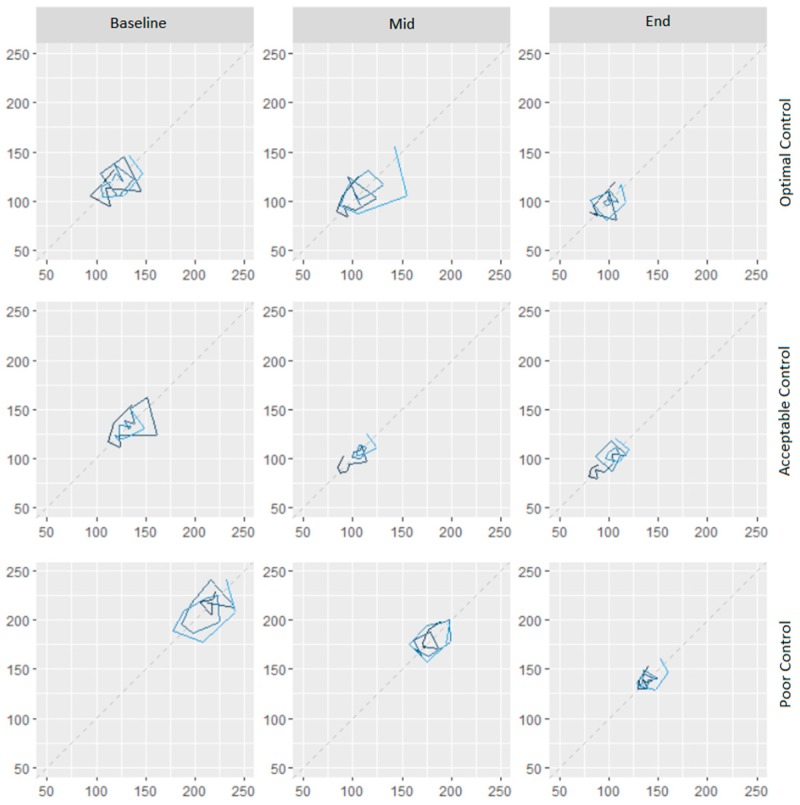
Composite assessment of glycemic control and variability based on raw glucose values at the same time on two sequential days (lag plot).

**Table 1 medsci-07-00052-t001:** Definitions of glycemic variability measures [5,11,12,13,15,16].

GV Measure	Formula		Interpretation
SD	$\sqrt{\frac{{\left(x_{i}-\bar{x}\;\right)}^{2}}{k-1}}$	where: *x_i_* = individual observation $\bar{x}$ = mean of observation *k* = number of observations	Traditional measure of dispersion; Measures short-term, within-day variability; Easy to compute, used very often
% CV	$\frac{s}{\bar{x}}$ × 100	where: *s* = standard deviation $\bar{x\;}$ = mean of observation	Traditional measure of dispersion, standardized for mean; Measures short-term, within-day variability; Easy to compute using mean and standard deviation
MAGE	$\sum \frac{\lambda }{n}$	if: λ > *υ* where: λ = each blood glucose increase or decrease n = number of observations *υ* = 1 SD of mean glucose for 24 hour period	Average of all glycemic excursions (except excursion having value <1 SD from mean glucose) in a 24 h time period; Captures short-term, within-day variability; Most commonly used
CONGA	$\sqrt{\frac{\sum_{t=t_{1}}^{t_{k*}}(D_{t}-\;{\bar{D}}^{2}}{k^{*}-1}}$	where: *k** = No. of observations where, there is an observation *m* mins ago *GR_t_* = glucose reading at time *t* *m* = *n* × 60 *D_t_* = *GR_t_* − *GR_t−m_* $\;\bar{D}=\frac{\sum_{t=t_{1}}^{t_{k*}}D_{t}}{k^{*}}$	Standard deviation of summated difference between current observation and previous observation; Captures short-term, within-day variability; Complex calculation, specifically developed for CGM
MODD	$\frac{\;\sum_{t=t_{1}}^{t_{k*}}\left|GR_{1}-GR_{t-\hat{t}}\right|\;}{k^{*}}$	where: $\hat{t}$ = 1440 (60 × 24); if reading taken every 1 min 96 (4 × 24); if reading taken every 15 min 24 (1 × 24); if reading taken every 60 min	24 h mean absolute differences between two values measured at the same timepoint; short-term, inter-day variation; Needs additional computation
HBGI	$\frac{1}{n}\overset{n}{\underset{t=1}{\sum }}rh\left(BGi\right)$	where: $f\left(BG\right)=1.509\;\times [\left(\log_{e}(BG){)}^{1.084}-5.381\right]\;\mathrm{if}\;\mathrm{BG}\;\mathrm{is}\;\mathrm{measured}\;\mathrm{in}\;mg/dL$ $f\left(BG\right)=1.509\;\times [\left(\log_{e}(18\;\times BG){)}^{1.084}-5.381\right]\;\mathrm{if}\;\mathrm{BG}\;\mathrm{is}\;\mathrm{measured}\;\mathrm{in}\;mmol/dL$ $r\;\left(BG\right)=10\;\times f{\left(BG\right)}^{2}\;$ rl (BG)=r (BG) if f(BG)>0 and 0 otherwise	Log transformation of glucose values; Captures risk for predicting severe glycaemia; Complex calculation, easy to interpret
LBGI	$\frac{1}{n}\overset{n}{\underset{t=1}{\sum }}rl\left(BGi\right)$	where: rl (BG)=r (BG) if f(BG)<0 and 0 otherwise	Log transformation of glucose values; Captures risk for predicting severe hyperglycaemia (HGBI); Complex calculation, easy to interpret

SD = standars deviation; % CV = Percent coefficient of Variation; MAGE = Mean Amplitude of Glycemic Excursions; CONGA = Continuous Overall Net Glycemic Action; MODD = Mean of Daily Differences; HBGI = High Blood Glucose Index; LBGI = Low Blood Glucose Index; CGM = Continuous Glucose Monitoring.

**Table 2 medsci-07-00052-t002:** Domains and underlying activity under diabetes self-management educational (DSME) program.

Domain	Activity	Days													
		1	2	3	4	5	6	7	8	9	10	11	12	13	14
Knowledge	Understanding diabetes	✓			✓										✓
	Understanding therapies		✓										✓		
	Improving self-care	✓			✓										
	Foot care								✓						
Physical activity	A 30-min brisk walk		✓	✓	✓	✓	✓	✓	✓	✓	✓	✓	✓	✓	✓
	Using activity trackers						✓	✓	✓	✓	✓	✓	✓	✓	✓
	Yoga/meditation			✓						✓				✓	
	Physical activity rewards						✓						✓		
Nutrition	Meal planning	✓		✓				✓			✓				
	Low-GI breakfast				✓	✓	✓	✓	✓	✓	✓	✓	✓	✓	✓
	Food diary feedback					✓			✓			✓			✓
	Meal planning rewards					✓						✓			
Behavior	Stress reduction		✓								✓				
	Coping skills						✓							✓	
	Tobacco cessation			✓						✓					
Disease management	CGM insertion/removal	✓													✓
	CGM readings and feedback		✓	✓	✓	✓	✓	✓	✓	✓	✓	✓	✓	✓	
	Screening for complications					✓									
	Drug prescription review							✓							✓
Activity count		4	4	5	5	6	6	6	6	6	6	6	6	6	7

**Table 3 medsci-07-00052-t003:** Baseline characteristics: Age, gender, years since diagnosis, complications, etc., by glycemic control status groups.

Baseline Characteristic	Optimal Control (*n* = 12)	Acceptable Control (*n* = 12)	Poor Control (*n* = 22)	Overall (*n* = 46)
	Mean (SD) or N (%)			
Male	8 (66.7%)	6 (50%)	10 (45.5%)	24 (52.2%)
Female	4 (33.3%)	6 (50%)	12 (54.5%)	22 (47.8%)
Age	56.7 (13.2)	54.5 (10.6)	52.2 (11.2)	54.0 (11.5)
BMI	26.5 (2.6)	25.8 (4.4)	26.5 (5.1)	26.3 (4.3)
Waist Circumference (*cm*)	97.8 (7.8)	93.4 (7.8)	97.7 (10.2)	96.59 (9.1)
Hip Circumference (*cm*)	100 (7.3)	99.3 (7.7)	103 (11.5)	101.4 (9.6)
WHR	0.98 (0.1)	0.94 (0.1)	0.95 (0.1)	0.96 (0.1)
SBP	126 (22)	130 (9.9)	132 (17.8)	129.8 (17.2)
DBP	82.5 (10.4)	77.8 (10.3)	83.4 (10.5)	81.7 (10.45)
Body Fat Percentage ^‡^	26 (4.1)	24.8 (5.1)	28.1 (4.2)	26.6 (4.6)
HbA1c	6.6 (0.3)	7.5 (0.3)	9.5 (1.2)	8.21 (1.6)
Duration of diabetes (years)	6.8 (8.8)	10.1 (6.9)	8.4 (7.8)	8.4 (7.8)
Hypertension	5 (41.7%)	7 (58.3%)	8 (36.4%)	20 (43.5%)
IHD	0 (0%)	0 (0%)	0 (0%)	0 (0%)
Hypothyroidism	3 (25.0%)	2 (16.7%)	1 (4.5%)	6 (13.0%)
Stroke	1 (8.3%)	0 (0%)	0 (0%)	1 (2.2%)
PVD	0 (0%)	0 (0%)	0 (0%)	0 (0%)
Retinopathy	0 (0%)	0 (0%)	1 (4.5%)	1 (2.2%)
Neuropathy	1 (8.3%)	2 (16.7%)	3 (13.6%)	6 (13.0%)
Nephropathy	1 (8.3%)	2 (16.7%)	0 (0%)	3 (6.5%)

BMI—body mass index; WHR—waist–hip ratio; SBP—systolic blood pressure; DBP—diastolic blood pressure; IHD—ischemic heart disease; PVD—peripheral vascular disease. ^‡^—Two missing from poor control group.

**Table 4 medsci-07-00052-t004:** Change in mean blood glucose and glycemic variability measures.

Measure	Baseline (Day 2)	Mid (Day 7)	End (Day 13)
**Optimal Control (*n* = 11)**			
Mean Glucose	115.90	107.96	98.21
SD Glucose	34.15	34.15	34.86
Coefficient of variation	29.46	31.64	35.49
MODD	24.56	24.44	23.14
MAGE	126.15	117.27	111.55
CONGA	8.67	8.24	7.61
HBGI	3.13	2.93	3.05
LBGI	2.30	3.57	5.95
**Acceptable Control (*n* = 10)**			
Mean Glucose	127.22	104.29	102.21
SD Glucose	50.02	27.62	25.50
Coefficient of variation	39.32	26.48	24.94
MODD	30.79	21.53	18.90
MAGE	138.01	109.43	105.65
CONGA	8.87	6.46	5.69
HBGI	5.56	1.41	1.34
LBGI	4.08	4.00	3.47
**Poor Control (*n* = 20)**			
Mean Glucose	203.67	176.81	144.62
SD Glucose	77.43	62.22	53.26
Coefficient of variation	38.02	35.19	36.83
MODD	45.82	37.04	32.28
MAGE	216.63	186.93	154.13
CONGA	14.10	9.62	8.38
HBGI	16.62	10.99	7.16
LBGI	3.41	2.14	4.06

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
