# Peer review of "Patterns of Glycemic Variability During a Diabetes Self-Management Educational Program"

_medsci, 2019, doi:10.3390/medsci7030052_

Round 1

Reviewer 1 Report

This is an interesting paper on glucose variability utilizing CGM in patients with type 2 diabetes.

In this study, glycemic variability was estimated by using various indices among patients of different control status along a diabetes education program.

The authors conclude that glycemic variability indices showed changes over a short period of time and thus could be used in clinical practice to assess glucose control status.

I would only suggest to standardize figures shown in Tables. Covariates age, glucose, DBP, SBP, pulse and height do not need decimals; and in covariates BMI and weight, one decimal is enough. 

Author Response

The Authors thank the Reviewer for his/her invaluable comments and  words of encouragement.

We have tried to clarify the points mentioned to the best of our ability and knowledge.

Thank You.

Reviewer 2 Report

Comments

This study estimated glycemic variability by using various indices among patients with optimal, acceptable and poorly controlled type 2 diabetes. The study very interesting and the results are very important. I have some comments for clarity.

Why almost half of the study participants refused?

The design of the study is “longitudinal-experimental” or “longitudinal intervention” instead of longitudinal study. It is a CGM plus educational interventions.

The intervention procedure needs a more detailed description of the intervention. A good description of interventions is important for replicating such interventions or identifying the contents of the interventions.

Although Table 1 is a good summary of the intervention contents, some of the intervention activities are not clear. For example:

1.       Tobacco cessation training was provided.  However, it is not clear whether this training was provided for all or only for smokers. This is difficult to assume as a reader because the smoking status of subjects was not also presented.

2.       Physical activity rewards were provided. What were the rewards?

3.       Activity trackers: it is not clear which activity trackers were used. Was it similar or any activity tracker the subject prefers?

It is not clear whether sensor data was transmitted from the user to the research centre every day via the internet, or whether it was centrally downloaded from the sensor after the completion of the intervention.

The authors reported no external funding was received. The authors need to report who sponsored the CGM sensors, that should also be explicitly acknowledged.  

In the procedures section (line 103), the authors mentioned data on serum creatinine, lipids and urine albumin to creatinine ratio were collected. However, none of these data were reported in the results section.

CMJE recommends that any form of intervention should be registered prospectively in a public trial registry database. If registered, please include registration number. 

The glycemic control was defined based on the baseline HbA1c values into optimal, acceptable and optimal control. Please add reference.

Line 141-142, please mention the type of R package used for producing the plots.

Footnote of Table 2 on page 5: the key with * is not needed because there is no data with that mark on the table.

Table 2 on page 6 should be Table 3. Check the caption.

The results in Table 6 would be more interesting if it has results of analytical statistics to indicate whether the changes for each glycemic variability measures between the three intervention endpoints was significant.

Line 157, change the word “associated” as it might be confused for a statistical association.

Figure 1 is too small. Either change the scale or consider dropping it. Much of the information is presented in Table 3.

Figure 2 needs further interpretation.

Data about average CGM readings for daily and all intervention endpoints is required for all groups.

Discussion: Why should the benchmarking be determined using the results at the intermediate point instead of at the endpoints of the intervention?

Author Response

(The authors gave the same response as above.)

Reviewer 3 Report

Authors conducted a longitudinal study with 2 weeks follow-up to investigate the effects of a DSME program on glycaemic variability. Glycaemic variability is an interesting topic and potentially a novel glycaemic target. While this manuscript has the potential to be of interest for the scientific community, there are some issues that need to be addressed.

Firstly, the authors firmly state that previous studies have reported GV as an independent risk factor for Association between GV and complications and mortality. Nevertheless, a recent review [Ceriello 2018, Lancet Diabetes Endocrinol], which is also cited by the authors, concluded: “Now that the improved availability of CGM has made blood glucose monitoring easier and more meaningful, glycaemic variability is emerging as an additional glycaemic target, even though doubt remains over whether both short-term or long-term glycaemic variability should be considered independent risk factors for diabetes-related complications.”. The authors should consider nuancing this in the introduction section.

Secondly, in the introduction section the authors refer to box 1 for the definitions of different GV measures. In my opinion, this box does not aid to clarity. The authors should consider reporting the definitions in a table rather than a box. They could for example add columns with the formula, the definition/interpretation and the ease to use/calculate in clinical practice.

In my opinion, the most important limitation of the manuscript is the unclear aim. Were the authors mainly interested in the effects of the DSME intervention on GV or were they more interested in the behavior of GV over time? What exactly were the research questions? Did the authors have any hypotheses? Depending on the actual aim of the authors, they may want to report less on the DMSE intervention.

Moreover, the authors do not mention whether they studied patients with type 1 , type 2 or both. I think this is essential information when studying diabetes.

Regarding the results, I personally would prefer to see Figure 1 as one figure, with multiple colours for the different GV measures. This would make it much easier to see the differences in GV behavior. In my opinion, the Poincare plots do not add anything to the manuscript, especially not since there is no reflection on these results in the discussion section.

As a clinician, I found it interesting to see that in those with optimal glycaemic control, the mean glucose declined over time, but this seems to be mainly caused by spending more time at low blood glucose values. Even though the LBGI is not a proxy for hypoglycaemic events, this makes me wonder whether it is safe for this subgroup to pursue stricter glycaemic control. The authors do not reflect on this aspect in their discussion section.

Please find some additional minor comments below.

Abstract:

Aim described in the abstract doesn’t match the aim in the main text

Please write the cut-off values for poor, acceptable and optimal control

For the results: consider to start with the number of patients included, followed by the number of CGM measures.

Introduction:

Line 41: consider interventions instead of approaches

Line 42: what do you mean advanced T2DM?

Line 43: throughout the manuscript there are some unnecessary spaces, such as here before [4]

Line 46: “From positive association …” unclear sentence, please reframe

Line 54: add “the” before “definition and interpretation”

Line 56: consider using “have become” instead of “became”

Lines 57-58: the measures GVP, MSE and PGS are reported here in the text, but are not found in the box, neither elsewhere in the manuscript. Consider omitting these measures

Consider using dual reporting for glucose, so both mg/dL as mmol/L

Methods:

Line 77-81: consider omitting “patients with diabetes … to promote self-management skills” since it doesn’t add anything to the manuscript

Line 87: why was living within a 5-10 km radius of the hospital one of the inclusion criteria?

Please refer to table 1 in the methods section

I found it hard to understand the Poincare plots and I do not see the additional value of them

Results:

Line 156: report the mean BG instead of the range

The authors have labeled two tables as “Table 2”

This also depends on the exact aim of the authors, but if the aim is to investigate the effectiveness of the DSME intervention on GV I would like to see some statistical tests and p-values reported in Table 2 [changes in blood glucose and GV]

Line 163: punctuation should be after “(Figure 1)”

Line 165: please remove punctuation between sub-groups and became

Line 166: I think that this definition of HBGI is more clear than the one in box 1.

Table 2 [change]: use 1 decimal

Table 2 [baseline]:

Please report n with their associated percentages

Many variables in this table report on approximately the same aspect. Therefor I suggest to omit weight and height and keep BMI, delete obesity. Additionally, omit hip circumference, WHR and body fat percentage.

In my opinion, pulse values do not add much

Consider reporting microvascular and macrovascular complications instead of the separate components such as retinopathy, neuropathy and nephropathy

There is inconsistency in the number of decimals reported. Report a maximum of 1 decimal when appropriate.

Discussion:

Line 194: the authors state that there is an absence of benchmarks. Isn’t target values a better word for this? Was this also one of the aims of the study, since in the discussion section the authors propose benchmarks based on their results?

Lines 205-215: the definition and explanation of the different GV measures should be in box 1, since it helps reader to understand the manuscript.

Line 226: consider “reflect increased risk for hypoglycaemic events” instead of “will reflect more chances of hypoglycaemic events”

Author Response

(The authors gave the same response as above.)

Reviewer 4 Report

Brief Summary

The aim of the study was to explore glucose variability (GV) among adults with type 2 diabetes who participated in a 2-week self-management education program. The study suggests that overall GV was reduced in the “poor control group” (A1C ≥ 8) and in the “acceptable control group” (A1C: 7.0-7.9) by the end of the 2-week program. The study is useful because it addresses the important issue of GV with respect to self-care in the T2DM population. Moreover, the study has value in that it examined the impacts of a comprehensive education program on self-management in this population.

Broad Comments

1.     Overall, the article should be revised to meet the clinical and scientific standards of the journal as discussed below. On a positive note, the visual aids were particularly helpful. Table 1 gives a good overview of the nature of the 2-week education program, and Figures 1 and 2 visually clarify the study results involving patterns of GV.

2.     The comparative part of the Results section (page 6 of 10), including the text and Table 3, should be more precise. For example, the authors mention only that “there was a fall in the mean BG values” or “a reduction in Glycemic variability indices”; however, specific values should be presented along with an indication of whether there is a statistically significant difference involved.

3.     In addition, a statistical analysis of the GV results (i.e., mean, SD, CV, MODD, MAGE, CONGA, HBGI, and LBGI)  should be performed using non-parametric repeated-measure ANOVA among the groups (optimal , acceptable, and poor control) and time periods (Baseline, Mid, and End). In other words, the authors should present information on whether the GV reductions were or were not statistically significant. In addition, Table 3 currently presents only mean GV values. After ANOVA is performed, both mean values and standard deviations for all variables should be presented in Table 3.

4.     In the Discussion section, this study’s GV results should be meaningfully compared to those of any previous studies involving T2DM self-management education programs.

5.     The article currently has many grammatical and other mechanical errors that distract the reader from the content. The entire article should be edited to eliminate these errors.

With these revisions and those identified below, the article has the potential to make a genuine contribution to the journal. In the Specific Comments, I pinpoint areas of concern in more detail and provide suggestions for improving the article.

Specific Comments

1.     The education program is described in a variety of ways as noted below. The exact name of the program should be consistently used throughout the article.

Line 2-3 (a diabetes self-management educational program),

Line 22-23 (a two-week self-management education program),

Line 68 (a two-week self-management education program),

Line 81 (a special two-week educational program),

Line 88 (a two-week intensive educational program),

Line 93 (a multi-modal self management educational program),

Line 111 (diabetes educational program),

Line 155 (2-week program),

Line 164 (2-week program),

Line 183 (an intensive diabetes educational program),

Line 199 (intensive educational program),

Line 203 (two-week educational program),

Line 233 (2-week educational program),

Line 238 (intensive educational program),

Line 242 (educational program)

2.     Line 72:

“Patients with diabetes mellitus” should be revised to “patients with type 2 diabetes mellitus”

3.     Line 84:

The Participants section indicates that study subjects had HbA1c values >7% but never actually specifies that they had been diagnosed with T2DM. The text should be clarified in this respect.

4.     Line 85:

The text states that “We identified adults (age more than 18 years).” However,

Table 2 (page 5 of 10) indicates that participants’ ages ranged from approximately 41 to 70 years.  This apparent inconsistency should be resolved in the article.

5.     Line 89:

The text is unclear about how HbA1c was measured in participants during a pre-screening visit. The text should be revised to specify the measurement method.

6.     Line 100- 101:

The text states that participants “performed an assessment of their quality of life (QoL).” It is unclear whether this assessment involved completing a questionnaire, and no questionnaire results are presented in the article. The text should specify how the QoL assessment was performed, and the assessment results should be briefly presented.

7.     Line 102:

The text states that fasting blood samples and urine samples were collected from participants on the fifth day of the program. However, the text does not explain what these samples were used for, and it is not clear why such samples were collected at approximately mid-program but apparently not at baseline and/or at the end of the program. The text should clarify these matters and should discuss whether the samples had any relationship to GV.

8.     Line 111 & 96-97:

The education program operated continuously for 14 days. The text does not specify whether all the participants were present on every day of the program without any absences. Also, if a participant was absent on a given day, it is not clear whether that participant was counted as having completed the program. The text should be revised to clarify these matters.

9.     Line 120:

In the Statistical Analysis section, each participant wore a continuous glucose monitor for 14 days. Because the device recorded a blood glucose level only every 15 minutes, there is a possibility of missing data. The text should be revised to clarify whether any data were missing, and if so, how they were managed.

10. Line 160:

Table 3 is misnumbered as Table 2. This error should be corrected.

Author Response

(The authors gave the same response as above.)

Round 2

Reviewer 2 Report

The authors satisfactorily answered my comments. The repeated measure ANOVA results table and the daily average CGM readings table were great additions. However, none of these tables are added in the revised version of the manuscript. I though they might be included in the Supp material. No, they are not there. Am I missing something here? 

In the ANOVA table, it is not clear whether the p-values are for the change in CVs or the change in mean HbA1c. It would also be interesting to see the significant changes in the other variability indices. 

Author Response

Dear Sir/Madam,

Greetings!

The authors are really thankful for all your valuable and constructive remarks and suggestions towards the manuscript.

Please find the Point-to-Point response to your queries and suggestions below.

Point 1: The authors satisfactorily answered my comments. The repeated measure ANOVA results table and the daily average CGM readings table were great additions. However, none of these tables are added in the revised version of the manuscript. I though they might be included in the Supp material. No, they are not there. Am I missing something here?

Response 1:

We thank you for your positive comment. We apologize for not including the repeated measures ANOVA tables and the daily average CGM readings table as the Supplementary Annexure. We have added the same in the Supplementary Annexure.

Point 2: In the ANOVA table, it is not clear whether the p-values are for the change in CVs or the change in mean HbA1c. It would also be interesting to see the significant changes in the other variability indices.

Response 2:

The p-values are for the change in the mean glucose readings for each group at different time-points (Baseline, Mid and End). The same has been mentioned in the Table provided in the Supplementary Appendix. The authors thank the reviewer for pointing this out.

The other glycemic variability indices reported in this study (MAGE, CONGA, MODD, HBGI, LGBI) are mathematically transformed and standardized estimates of standard deviation which may not fulfill the necessarily prerequisites (the assumptions of normality and sphericity) required for repeated measures ANOVA. Moreover, these GV measures philosophically offer the information pertaining to the magnitude of the change in the variance in reference to time.

Reviewer 3 Report

Authors have improved the manuscript well enough.  I have two minor comments:

Line 49-51: I miss a verb in this sentence

There is no referral to Table 2 in the tekst

Author Response

Dear Sir/Madam,

Greetings!

The authors extend their sincere thanks for your support for the betterment of the manuscript.

As suggested, we have made the following changes:

Point 1: Line 49-51: I miss a verb in this sentence

Response 1:

The sentence has been re-written as follows:

The second caveat is related with the extent of capturing the true the mean glucose value by 7-point glucose estimation method.

Point 2: There is no referral to Table 2 in the tekst

Response 2: We apologize for this oversight. We have referred the Table 2 at the appropriate place in the manuscript. (Line 114)